# Television watching and cognitive outcomes in adults and older adults: A systematic review and dose-response meta-analysis of observational studies

Hattapark Dejakaisaya[1], Wiriya Mahikul[1], Nat Na-ek[2,3,4], Chanawee Hirunpattarasilp[1,5*]

1 Princess Srisavangavadhana Faculty of Medicine, Chulabhorn Royal Academy, Bangkok, Thailand,
2 Division of Pharmacy Practice, Department of Pharmaceutical Care, School of Pharmaceutical Sciences, University of Phayao, Phayao, Thailand, 3 Pharmacoepidemiology, Social and Administrative Pharmacy (P-SAP) Research Unit, School of Pharmaceutical Sciences, University of Phayao, Phayao, Thailand, 4 Unit of Excellence on Cardiovascular Archive Research and Clinical Epidemiology, School of Pharmaceutical Sciences, University of Phayao, Phayao, Thailand, 5 Department of Neurology, University Hospitals Cleveland Medical Center, Cleveland, Ohio, United States of America

* chanawee.hir@cra.ac.th

## Abstract

This systematic review and meta-analysis aimed to examine the association between television watching and cognitive outcomes in adults and older adults as the current evidence is inconsistent. We searched the Cochrane, MEDLINE, Embase, PsycINFO, Scopus, and Web of Science databases for relevant studies from inception to June 30, 2024. Risk of bias was assessed using the Newcastle–Ottawa Scale. Dose–response and conventional meta-analyses were performed using one-stage random-effects and DerSimonian and Laird random-effects models, respectively. Our systematic review included 35 studies with 1,292,052 participants (8,572 cases of cognitive impairment), of which 28 studies were further meta-analyzed. A dose–response meta-analysis revealed a nonlinear association between time spent watching TV and an increased risk of cognitive impairment (Wald test p-value = 0.04), particularly for viewing durations of ≥4 hours per day. Additionally, watching ≥6 hours of television per day was associated with a significant decrease in cognitive score (standardized beta coefficient = −0.09; 95% CI: −0.17, −0.003; $I^2$ = 71.8%; seven studies). Also, a longer television-watching time was associated with a lower cognitive score (pooled standardized mean difference = −0.02; 95% CI: −0.03, −0.003; $I^2$ = 66.45%; six studies). Watching television for a longer period was associated with negative cognitive outcomes in adults and older adults. Further research is needed to confirm this association and elucidate the underlying biological mechanisms.

**Data availability statement:** All relevant data are within the manuscript and its Supporting Information files.

**Funding:** This research was supported by Chulabhorn Royal Academy and University of Phayao and Thailand Science Research and Innovation Fund (Fundamental Fund 2024). There was no additional external funding received for this study. The funders had no role in study design, data collection and analysis, decision to publish, or preparation of the manuscript.

**Competing interests:** The authors have declared that no competing interests exist.

## Introduction

The global trend toward aging of the population has increased the prevalence of diseases associated with aging. One of these diseases is dementia, a syndrome with various etiologies causing a decline in cognitive abilities that interferes with activities of daily living, leading to functional impairment. It is the seventh leading cause of death and a major cause of disability and dependency among older adults, according to the World Health Organization [1]. Moreover, the number of people with dementia is expected to increase from 55 million in 2019–139 million in 2050 [2]. This will in turn increase the burden imposed by dementia on the global healthcare system, doubling the associated cost from US$1.3 trillion in 2019 to US$2.8 trillion by 2030 [3].

There are more than 100 causes of dementia [4]; the most common one is Alzheimer's disease (AD), accounting for ≥50% of all cases [3]. AD causes a progressive deterioration in two or more cognitive domains, especially episodic memory and executive functions [5], causing patients to suffer from symptoms such as memory loss and spatial disorientation [6]. In addition, AD may enhance the mortality rate by up to 40% [7] because of complications related to aspiration, infection, or inanition [8]. AD can be caused by a myriad of pathological changes in the brain, such as the accumulation of certain amyloid-β peptides [9], neurofibrillary tangles [10], dysfunctional glutamatergic pathways [11], and vascular changes [12]. While a disease-modifying therapy, lecanemab, is available, it only slows the progression of mild AD and is not a curative treatment [13]. Similarly, other types of dementia, such as frontotemporal dementia, dementia with Lewy bodies, and vascular dementia, lack disease-modifying therapies. As a result, dementia remains incurable [3]. Therefore, risk mitigation remains the most effective strategy to address the global rise in dementia cases.

Understanding how activities of daily living in adults and older adults affect the risk of developing dementia may provide insights into how the global population can age in a healthier way. Therefore, it is imperative that any positive or negative impact of common daily leisure activities on cognition is identified. Among these common daily leisure activities, television (TV) watching is of particular interest as it is one of the most popular leisure activities among adults and older adults [14]. TV watching duration is widely measured to indicate the amount of sedentary behavior a person engages in and, currently, longer TV watching durations are considered to be related to an elevated risk of obesity [15–18], type 2 diabetes [19,20], and cardiovascular disease [21,22].

Despite this, there is still no consensus on the impact of TV watching on cognition because there is evidence supporting both positive [23,24] and negative [25,26] impacts. This discrepancy may partly be explained by differences in study designs and methodological aspects. Furthermore, no previous studies have examined the association between TV watching time and the risk of cognitive impairment as a nonlinear function. The current investigation is therefore warranted, and the aim of this study is to establish whether there is a relationship between TV-watching time and cognitive outcomes in adults and older adults. Performing a systematic review and meta-analysis allows us to examine the impact of methodological differences on

the observed association. Additionally, conducting a dose-response meta-analysis enables us to investigate the nonlinear relationship between TV watching time and cognitive outcomes.

## Materials and methods

This report followed the Preferred Reporting Items for Systematic Reviews and Meta-Analysis (PRISMA) guidelines [27], and the protocol was prospectively registered on PROSPERO (CRD42023408255). Our PRISMA checklist is shown in S1 Table. Further, this project received an ethics exemption from Chulabhorn Royal Academy's ethics committee (project number EC 052/2566).

### Search strategy

Six databases (the Cochrane, Ovid MEDLINE, Ovid Embase, PsycINFO, Scopus, and Web of Science databases) were searched from inception until June 30, 2024. We searched for articles using the keywords "television," "cognitive function," "neuropsychological test," "dementia," "elderly," and "adult." Details of the search strategy used for each database can be found in the S1 file. Additional studies were also identified through a manual search of reference lists.

### Study selection and eligibility criteria

We considered all abstracts and publications, with no restrictions on date or language. For inclusion in our meta-analysis, the adults and older adults (≥18 years old) in each study had to be unaffected by serious disability such as visual impairment, auditory impairment, cognitive impairment, or dementia (at the start of the study), and to not be taking drugs that affect cognition. Furthermore, the interventions in the studies had to not involve special types of TV-watching regimens (e.g., TV-based cognitive training programs). All identified records were screened independently by two reviewers (HD, CH): first, the titles and abstracts were screened, followed by the full texts, and relevant information was independently extracted. Any disagreements between the two reviewers were resolved through discussion or by a third reviewer (WM, NN) if necessary.

### Data extraction

We collected data on all cognitive outcomes from individual studies, including cognitive scores on standardized tests and risk data for mild cognitive impairment (MCI) and dementia. Further criteria applied when extracting data from studies with overlapping populations, multiple levels of TV viewing, or multiple cognitive outcomes can be found in S1 file.

Studies with overlapping populations or from the same database were ranked based on a designed hierarchy and the studies with the highest hierarchical score were included. Briefly, studies were ranked based on 1) the most relevant outcomes (e.g., dementia, cognitive impairment, and cognitive score); 2) sample size (largest); and 3) year of publication (latest), respectively. For articles with multiple levels of television viewing, all data were collected to analyze the dose-response relationship. For studies that report both cognitive score and MCI/dementia risk, we collected both outcomes for their respective meta-analyses.

### Assessment of bias of individual studies

The Newcastle–Ottawa Quality Assessment Scale (NOS) was applied to evaluate and analyze the methodological quality of each study [28,29]. Three domains were evaluated: selection, comparability, and outcome/exposure assessment. Two reviewers (HD, WM) independently scored studies as low (8–9 points), moderate (6–7 points), or high (0–5 points) risk of bias. Disagreements were resolved by discussion. A summary of risk levels and visualizations was generated using robvis [30]. The standard NOS was used for case–control or cohort studies, whereas a modified scale [31] was used for cross-sectional studies. Details of the assessment and risk stratification performed using the NOS can be found in

S1 file. No studies were excluded based on the bias assessment; however, a sensitivity analysis of only studies with a low-to-moderate risk of bias was performed.

## Statistical analysis

**Data items.** Data on the following aspects were extracted from each study: 1) demographics; 2) characteristics of the study population; 3) TV-watching quantification methods; and 4) cognitive outcomes. Detailed data are listed in S1 file. In studies with multiple levels of TV exposure, we used the reported mean or median to determine the dose (time) of TV watching in each exposure category; otherwise, range values were converted to specific doses according to the method suggested by Shim et al. [32]. Additionally, the standard error and standard deviation of outcomes were derived from the upper and lower limits of the 95% confidence interval (CI) using standard formulas [33,34].

**Synthesis methods.** There were two main outcomes in this study: risk of cognitive impairment (i.e., MCI, dementia, or AD) and cognitive score. To perform a dose–response meta-analysis, a one-stage random-effects model was used [32]. In brief, we initially created a scatter plot of each outcome (y-axis) and the TV-watching time (x-axis) to visualize the crude association. Then, a linear regression model was fitted. To examine nonlinear associations, we fitted the model with either a quadratic term or a restricted cubic spline with three, four, or five knots, where the location of each knot was specified according to the recommended percentile position [35]. In addition, we examined nonlinearity with the Wald test. Lastly, we selected the best-fitted model, i.e., with the lowest Akaike information criterion (AIC) or Bayesian information criterion (BIC). The binary outcome (cognitive impairment risk) was analyzed using the Greenland and Longnecker method and a restricted maximum-likelihood random-effects model, whereas for the continuous outcome (cognitive score), dose–response meta-analysis was performed using Cohen's standardized mean difference approach. To avoid duplication issues, we analyzed only one outcome from each study that reported more than one outcome with the same type of variable (continuous or binary), using the following hierarchy: 1) If both cohort and cross-sectional results were reported [36], we used the cohort results; 2) If each outcome was reported along with a cumulative one [37], we selected the cumulative one; 3) If both short- and long-term outcomes were reported [23], we chose the long-term one; 4) If each outcome was reported separately without a cumulative one [38,39], we used the outcome with the smallest variance.

In the conventional meta-analysis, the risk of cognitive impairment was given by a ratio effect size or mean difference, using the shortest TV-watching-time group as the reference group. In contrast, the cognitive score was indicated by a beta coefficient derived from the regression model. Where possible, we used the effect sizes from models that included the most comprehensive set of covariates reported in each primary study to account for potential cofounders such as age, sex, education, socioeconomic status, lifestyle factors (e.g., physical activity, smoking), and other comorbidities. These factors, such as older age, female sex and lower education attainment, can negatively affect the cognitive outcomes [40]. Details of covariate adjusted for in each study are provided in Table 1. Because none of the studies reported prevalence data for the outcomes in the reference group or the absolute number of participants experiencing the outcomes in each group, we could not convert hazard ratios to odds ratios (or vice versa). Consequently, all ratio effect sizes (i.e., risk ratio, odds ratio, and hazard ratio) were pooled in the main analysis, primarily using the DerSimonian and Laird random-effects model, and labeled as relative risk. Additionally, we performed subgroup analysis according to study design, type of outcome, risk of bias, and reported effect size.

**Reporting bias assessment.** To assess the statistical heterogeneity in the meta-analysis, we calculated the $I^2$ statistic (indicating the extent to which variance is explained by between-study heterogeneity) and the p-value for Cochrane's Q statistic. An $I^2 > 50\%$ and a p-value for the Q test <0.1 were taken to indicate a significant degree of heterogeneity. Subsequently, we sought to identify the source of heterogeneity by conducting subgroup analyses, with the subgroup having the smallest $I^2$ value likely being the source of heterogeneity [33].

Table 1. The main characteristics of the 35 studies included in the systematic review categorized by their study designs.

| 1st Author, Country, Year [ref] | Sample size | Male (%) | Mean age at baseline (years) | Mean follow-up (years) | TV viewing (CAT/CON) | Outcome | Type of cognitive test/diagnostic method | Key Finding: Cognitive impairment risk | Key Finding: Cognitive score | Adjusted confounder(s) |
|---|---|---|---|---|---|---|---|---|---|---|
| Cross-sectional | | | | | | | | | | |
| Bakrania, UK, 2018 [42] | 502,643 (UK Biobank) | 45.60 | 56.50 | N/A | CAT | Cognitive score | Computerized cognitive test questionnaire | N/A | N/A | Age, sex, BMI, smoking status, alcohol consumption, ethnicity, employment status, socioeconomic status (household income and education), disability/illness, total physical activity level, fruit and vegetable consumption, sleep duration |
| Chen, Japan, 2021 [43] | 3,708 | 50.00 | 49.49 | N/A | CAT | Dementia | Exclusive checklist for the dementia diagnosis | N/A | N/A | Age, sex, marital status, education level, employment status, diabetes diagnosis |
| Coelho, Canada, 2020 [44] | 75 | 20.00 | 75.60 | N/A | CON | Cognitive score | BRIEF-A | N/A | N/A | Age, physical activity (leisure score index) |
| Da Ronch, Italy, 2015 [45] | 1217 (MentDis_ICF65+) | 52.40 | 73.14 | N/A | CON | Cognitive score | MMSE | N/A | N/A | Age, sex, education level, GDS, no. of chronic diseases, use of psychoactive medications, living situation (alone vs. with someone), ADL |
| Heisz, Canada, 2015 [46] | 61 (Younger: 31, Older: 30) | Younger: 48.00, Older: 50.00 | Younger: 24.00, Older: 74.00 | N/A | CAT | Cognitive score | Face recognition paradigm | N/A | N/A | Age (younger vs. older adults), Physical activity level |
| Jopp, USA, 2007 [47] | 326 | 38.00 | 55.48 | N/A | CAT | Cognitive score | Multiple standardized cognitive tests | N/A | N/A | Age, sex, education level, self-rated health, functional ability, no. of chronic conditions |
| Jung, South Korea, 2020 [25] | 336 (NSOK) | 33.00 | 71.20 | N/A | CAT | MCI | MMSE | N/A MA: ⊕ | N/A | Age, sex, marital status, education level, living arrangement, income, health conditions, depressive symptoms, health behaviors (smoking, alcohol), physical activity |
| Mellow, Australia, 2022 [48] | 384 (ACTIVate) | 31.50 | 65.50 | N/A | CAT | Cognitive score | ACE-III | N/A | DR: ① N/A | Age, sex, education level, BMI, employment status, depression symptoms, sleep quality, chronic conditions |
| Olanrewaju, UK, 2020 [36]* | 6,395 (TILDA) | N/A | >50.00 | N/A | CON | Cognitive score | MMSE | N/A | N/A | Age, sex, education level, wealth index, marital status, employment status, physical activity, mental health status |
| Ringin, UK, 2023 [49] | 59,653 (UK Biobank) | 52.10 | 56.99 | N/A | CON | Cognitive score | Standardized cognitive tests | N/A | N/A MA: ① | Age, sex, SES, education, alcohol, smoking, physical activity, computer use, waist circumference, sleep duration, diagnostic group (bipolar vs healthy) and their interactions |
| Rosenberg, USA, 2016 [37] | 307 | 27.70 | 83.60 | N/A | CON | Cognitive score | Trail Making Test A & B | N/A | N/A MA: 0 | Age, sex, education, marital status, and physical activity |

*(Continued)*

Table 1. (Continued)

| 1st Author, Country, Year [ref] | Sample size | Male (%) | Mean age at baseline (years) | Mean follow-up (years) | TV viewing (CAT/CON) | Outcome | Type of cognitive test/diagnostic method | Key Finding — Cognitive impairment risk | Key Finding — Cognitive score | Adjusted confounder(s) |
|---|---|---|---|---|---|---|---|---|---|---|
| Tantanokit, Thailand, 2021 [50] | 295 | 21.00 | 70.23 | N/A | CAT | Cognitive score | MMSE | N/A MA: 0 | N/A | Age, education, mobile phone access, computer skill, internet skill, Family history of dementia |
| Wanders, Netherlands, 2021 [51] | 2,237 (NES) | 56.80 | 61.00 | N/A | CAT | Cognitive score | COST-A | N/A | DR: ⊕ N/A | Age, sex, education, Body Mass Index (BMI), working, smoking, alcohol consumption, health status, comorbidities/polypharmacy, sleep disturbances, Geriatric Depression Scale, Physical activity (measured in MET-minutes/week), Other sedentary domains (added in Model 4 for analyses of specific sedentary behaviors) |
| Yuan, China 2018 [52] | 2,617 | 54.03 | 69.06 | N/A | CAT | Cognitive score | MoCA | N/A | DR: ⊕ N/A | Age, sex, education, area, marital status, BMI, hypertension, diabetes, and depression |
| Cohort | | | | | | | | | | |
| Allen, UK, 2019 [38] | 9,551 (ELSA) | 45.60 | 67.42 | N/A | CON | Cognitive score | Standardized cognitive tests | N/A | N/A MA: ⊕ | Age, sex, race, education, alcohol, physical activity, diet, previous memory performance |
| Fajersztajn, Brazil, 2021 [39] | 1,243 (SPAH) | 38.60 | 71.70 | 2.00 | CON | Cognitive score, MCI, Dementia | CSI-D, CERAD, DSM-IV | N/A MA: 0 | N/A | Age, sex, education, occupation, income, functional status, baseline global cognitive function, and baseline amnestic mild cognitive impairment score |
| Fancourt, UK, 2019 [53] | 3,590 (ELSA) | 43.70 | 67.10 | 6.00 | CAT | Cognitive score | Standardized cognitive tests | N/A | DR: ⊕ N/A | Age, sex, race, marital status, education, employment, retirement, wealth, social support, depression, alcohol, smoking, physical activity, reading daily newspaper, internet use, self-reported physical health, chronic conditions, mobility problems, and baseline cognition. |
| Floud, UK, 2019 [23] | 645,967 (Million woman study) | 0.00 | 60.00 | Activity: 5.00 Cognition: 4.00 | CAT | Dementia | N/A | N/A MA: ⊕ | N/A | Education, marital status, employment, area deprivation, Townsen score, alcohol, BMI, smoking, physical activity, self-rated health, use of menopausal hormones, current treatment for depression, diabetes, and hypertension |
| Hamer, UK, 2014 [54] | 6,359 (ELSA) | 45.20 | 64.90 | 2.00 | CAT | Cognitive score | Standardized cognitive tests | N/A | DR: ⊕ N/A | Age, sex, social class, alcohol, BMI, smoking, physical activity, use of internet, reading daily newspaper, disability, chronic illness, baseline CES-D score, and interaction term (daily TV viewing and time) |

(Continued)

| 1st Author, Country, Year [ref] | Sample size | Male (%) | Mean age at baseline (years) | Mean follow-up (years) | TV viewing (CAT/CON) | Outcome | Type of cognitive test/diagnostic method | Key Finding Cognitive impairment risk | Key Finding Cognitive score | Adjusted confounder(s) |
|---|---|---|---|---|---|---|---|---|---|---|
| Hoang, USA, 2016 [55] | 3,247 (CARDIA study) | 43.50 | 25.10 | 25.00 | CAT | Cognitive score | DSST, Stroop, RAVLT | DR: ⊕ N/A | N/A | Age, sex, race, education, alcohol, BMI, smoking, and hypertension |
| Kesse-Guyot, France, 2012 [56] | 2,579 (SU.VI.MAX) | 55.30 | 65.60 | N/A | CAT | Cognitive score | RI-48, TMT | N/A | N/A | Age, sex, education, supplementation group, occupation, retirement status, BMI, smoking, physical activity, reading, computer use, CES-D score, general health status, and history of cardiovascular diseases, diabetes, and hypertension |
| Lin, Chinese Taipei, 2022 [57] | 4,440 (TLSA) | 53.10 | 68.89 | 12.00 | CAT | Cognitive score | SPMSQ | N/A | N/A | Age, sex, education, marital status, annual household income, living arrangement, occupation, residence, satisfaction with one's economic status, and the number of chronic diseases (hypertension, diabetes, heart disease, and stroke) |
| Maasakkers, Ireland, 2021 [58] | 1,276 (TILDA) | 47.00 | 67.30 | 8.00 | CON | Cognitive score | MMSE | N/A | N/A MA: 0 | Age, sex, education, marital status, alcohol, BMI, smoking, physical activity, sleep quality, perceived health status, blood pressure, depression, mobility limitations, and morbidities |
| Major, USA, 2023 [59] | 1,261 (Health ABC) | 47.80 | 75.10 | N/A | CAT | Cognitive score | 3MS, DSST | N/A | DR: ① N/A | Age, sex, race, education, physical activity, depressive symptoms (CES-D score), health status (self-rated health) |
| Nemoto, Japan, 2022 [60] | 5,323 | 45.50 | 74.70 | 5.00 | CAT | Dementia | Standardized cognitive tests | DR: ⊕ N/A | N/A | Age, sex, education, marital status, living status, employment status, health status (self-rated health), BMI, physical activity, reading time, medical treatment (stroke, diabetes, hypertension), and frailty |
| Palta, USA, 2020 [61] | 10,700 (ARIC) | 44.00 | 59.00 | 17.40 | CAT | Cognitive score | Standardized cognitive tests | N/A | N/A | Age, sex, race, education, income, neighborhood SES, BMI, smoking, diabetes, hypertension, and APOE ε4 |
| Raichlen, UK, 2022 [62] | 146,651 (Dementia: 3,507, No Dementia: 143,144, UK Biobank) | 40.46 (Dementia: 57.00, No Dementia: 50.30) | 64.59 (Dementia: 66.17, No Dementia: 64.55) | 11.87 | CAT | Dementia | Hospital record | N/A MA: ⊕ | N/A | Age, sex, race, Townsend deprivation index, education, alcohol, BMI, smoking, physical activity, computer use, sleep, healthy diet score, chronic disease, APOE ε4, depression, and social contact |
| Shin, USA, 2021 [63] | 3,793 (HRS) | 44.00 | 73.01 | N/A | CON | Cognitive score | Standardized cognitive tests | N/A | N/A MA: 0 | Sex, education level, age, marital status, employment, household income, net worth, health insurance ownership, number of children, self-reported health status, diagnoses of medical conditions, depressive symptoms, number of difficulties performing ADL and IADL, smoking, weight status, number of alcoholic drinks |

*(Continued)*

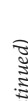

**Table 1.** (Continued)

| 1st Author, Country, Year [ref] | Sample size | Male (%) | Mean age at baseline (years) | Mean follow-up (years) | TV viewing (CAT/CON) | Outcome | Type of cognitive test/diagnostic method | Key Finding — Cognitive impairment risk | Key Finding — Cognitive score | Adjusted confounder(s) |
|---|---|---|---|---|---|---|---|---|---|---|
| Takeuchi, UK, 2023 [64] | 373,345 (Dementia: 4,086, No Dementia: 369,259, UK Biobank) | 47.10 (Dementia: 57.00, No Dementia: 47.00) | 66.87 (Dementia: 63.97, No Dementia: 55.78) | N/A | CAT | Dementia | Hospital record | DR: ⊕ MA: 0 | N/A | Sex, age, neighborhood-level socioeconomic status, education level, household income, current employment status, metabolic equivalent of task hours, number in household, body mass index, self-report health status, sleep duration, length of TV viewing at the first assessment visit, visuospatial memory performance at the first assessment visit |
| Wang, China, 2006 [65] | 5,437 (CI: 593, No CI: 4,844) | 51.68 (CI: 42.50, No CI: 52.80) | 63.42 (CI: 68.50, No CI: 62.80) | 4.70 | CON | MCI | MMSE | N/A MA: 0 | N/A | Age, sex, education, occupation, medical conditions, smoking, drinking, depressive symptoms, baseline MMSE, and ADL scores, and participation in other activities |
| Zhang, China, 2023 [66] | 2002=5,246 2005=5,138 2008=4,968 2011=3,504 2014=2,056 (CLHLS) | 2002=50.90 2005=49.60 2008=50.60 2011=51.70 2014=51.70 | Changes every year (categorical) | 16.00 | CAT | MCI | MMSE | N/A MA: ⊖ | N/A | age, sex, education, rural/urban residence, region, marital status dummy variables, and number of chronic diseases, family income per capita, number of living children |
| Olanrewaju, UK, 2020 [36]* | 5,655 (TILDA) | N/A | >50.00 | 2.00 | CAT & CON | Cognitive score | MMSE | N/A | DR: ⊖ MA: 0 | Age, sex, social class, employment, social participation, obesity, smoking, physical activity, loneliness, disability, depression, and chronic conditions |
| Shi, USA, 2024 [67] | 45,176 (NHS) | 0.00 | 59.20 | 20.00 | CAT & CON | Cognitive score | Structured Telephone Interview for Dementia Assessment | DR: ⊕ N/A | N/A | Age, education, marital status, annual household income, family history of cancer, myocardial infarction, and diabetes, baseline hypertension and high cholesterol, menopausal status and postmenopausal hormone use, aspirin use, smoking history, alcohol intake, total energy intake, and diet quality, sleep duration |
| **Case-control** | | | | | | | | | | |
| Lindstrom, USA, 2005 [26] | 446 AD: 135 Control: 331 | AD: 46.67 Control: 39.88 | AD: 83.00 Control: 81.00 | N/A | CON | Dementia | Hospital record | N/A MA: ⊕ | N/A | Year of birth, sex, income, and years of completed education |

*(Continued)*

**Table 1.** (Continued)

| 1st Author, Country, Year [ref] | Sample size | Male (%) | Mean age at baseline (years) | Mean follow-up (years) | TV viewing (CAT/CON) | Outcome | Type of cognitive test/diagnostic method | Key Finding: Cognitive impairment risk | Key Finding: Cognitive score | Adjusted confounder(s) |
|---|---|---|---|---|---|---|---|---|---|---|
| Ramos, Spain, 2021 [68] | 497 (CI: 153, No CI: 344) | 26.60 | 50–59: 74.00 60–69: 155.00 70–79: 191.00 ≥ 80: 75.00 | N/A | CON | MCI | MIS, SPMSQ, SVF | N/A MA: 0 | N/A | Age, subjective memory complaint, educational level, marital status, night-time sleep, reading time, internet and mobile device use |
| Zhao, China, 2015 [24] | 404 (control: 306, MCI: 98) | Control: 48.30 MCI: 50.00 | Control: 71.20 MCI: 84.63 | N/A | CON | MCI | MoCA | N/A MA: ⊖ | N/A | Age, sex, educational level, chronic disease (including hypertension, diabetes, coronary heart disease and cerebrovascular disease), body mass index (BMI), fasting plasma glucose (GLU), total cholesterol (TC), triglycerides (TG), high-density lipoprotein (HDL), low-density lipoprotein (LDL), alanine transaminase (ALT) and aspartate aminotransferase (AST). |

**Abbreviations:** ACE-III, Addenbrooke's cognitive examination III; ADL, activities of daily living; BMI, body mass index; CARDIA, coronary artery risk development in young adults; CAT, categorical; CERAD, consortium to establish a registry for Alzheimer's disease; CLHLS, Chinese longitudinal healthy longevity surveys; CON, continuous; COST-A, cognitive online self-test Amsterdam; CSI-D, community screening instrument for dementia; DR, dose-response; DSM-IV, the diagnostic and statistical manual of mental disorders-fourth edition; DSST, digit symbol substitution test; ELSA, english longitudinal study of aging; GSD, geriatric depression scale; HRS, health and retirement study; MA, meta-analysis; MCI, mild cognitive impairment; MIS, memory impairment screen; MMSE, mini mental state exam; MoCA, montreal cognitive assessment; NES, nijmegen exercise study; NSOK, national survey of older Koreans; N/A, not available; RAVLT, rey auditory verbal learning test; SES, socioeconomic status; SPAH, São Paulo aging & health study; SPMSQ, Spanish version of short portable mental state questionnaire; SVF, semantic verbal fluency; TILDA, the Irish longitudinal study on ageing; UK, United Kingdom.

**Note:** *The study consists of two or more study designs, and each design has been separately analyzed.

Key findings: 0 = null findings, ⊕ = positive association (i.e., increased TV watching time is associated with increased cognitive impairment risk or increased cognitive score), ⊖ = negative association (i.e., increased TV watching time is associated with decreased cognitive impairment risk or decreased cognitive score).

The potential for publication bias in the included studies was evaluated by creating a funnel plot of outcome versus the inverse of the standard error and conducting Egger's test (for datasets with more than 10 studies) [33]. A lack of asymmetry in the funnel plot and an Egger's test p-value >0.05 suggest that there is no evidence of publication bias.

**Certainty assessment.** We conducted the sensitivity analysis as follows: 1) we analyzed the data with a restricted maximum likelihood random-effects model; 2) we included only fully adjusted effect sizes; 3) we replaced the outcome with the largest variance; 4) we replaced the outcome with the shortest follow-up time; and 5) we used the Tweedie trim-and-fill method to adjust for potential publication bias. Additionally, the influence of each study was examined by performing a leave-one-out analysis. To evaluate the level of evidence for each outcome, we applied the Grading of Recommendations, Assessment, Development and Evaluation (GRADE) approach [41].

The dose–response meta-analysis was conducted using the "dosresmeta" package in the R program (version 4.3.1; R Development Core Team, Vienna, Austria), and conventional meta-analysis was performed using STATA software (version 16.1; StataCorp LLC, College Station, TX, USA).

## Results

### Screening results

Of the 7,363 studies initially screened, 35 were included in our systematic review, and 28 were included in the meta-analysis. Among these latter studies, 10 were cross-sectional, 15 were cohort, and 3 were case–control studies. In total, our study analyzed data from 1,292,052 participants, including 8,572 individuals diagnosed with cognitive impairment (135 with Alzheimer's disease, 8,339 with dementia, and 98 with mild cognitive impairment [MCI]). The PRISMA flow diagram is presented in Fig 1.

### Study characteristics

The characteristics of all studies included in the systematic review are shown in Table 1, and the risk of bias assessments for individual studies can be found in S1-3 Fig. In brief, 10 studies had a low risk of bias, 16 had a moderate risk, and 9 had a high risk.

**Analysis of the risk of cognitive impairment.** In the dose–response meta-analysis, we identified the 3-knot restricted cubic spline (RCS) model as the best-fitting model (S2 Table). This model demonstrated a nonlinear increase in the risk of cognitive impairment with longer TV-watching time (Wald test p-value = 0.04), particularly beyond 4 hours per day. Predicted relative risks (RRs) and 95% confidence intervals (CIs) for selected doses (hours of TV-watching time per day) are presented in S3 Table and visualized in Fig 2A. The initial scatter plot illustrating the association between TV-watching time and the risk of cognitive impairment is shown in S4 Fig to demonstrate statistical analysis transparency.

In the conventional meta-analysis, we did not find an association between longer TV-watching time and the risk of cognitive impairment: the pooled relative risk was 1.01 (95% CI: 0.95, 1.08; 11 studies; Fig 2B). Of note, there was a high degree of heterogeneity ($I^2 = 90.54\%$, $p < 0.001$), and the study design, type of outcome, reported effect size, and risk of bias of individual studies were not major sources of heterogeneity. Substantial heterogeneity was also detected in several analyses ($I^2$ ranging from 66.4% to 91.5%). Despite conducting subgroup and sensitivity analyses by study design, risk of bias, and outcome type, the source of heterogeneity could not be fully explained (see S4 Table and S5 Fig). Interestingly, we observed a significant association between longer TV-watching time and AD (odds ratio = 1.32 [95% CI: 1.08, 1.62; one study]; Fig 2B), and when combining only hazard ratios in subgroup analysis (pooled hazard ratio = 1.07 [95% CI: 1.02, 1.13; four studies]; Table 2, S4 Table). All sensitivity analyses showed similar null findings (Table 2). The results from the subgroup analysis based on study design are shown in S5 Fig.

**Analysis of the cognitive score.** Regarding the association between TV-watching time and cognitive score, we found a nonlinear relationship via a three-knot restricted cubic spline model, which was the best-fitting model in the

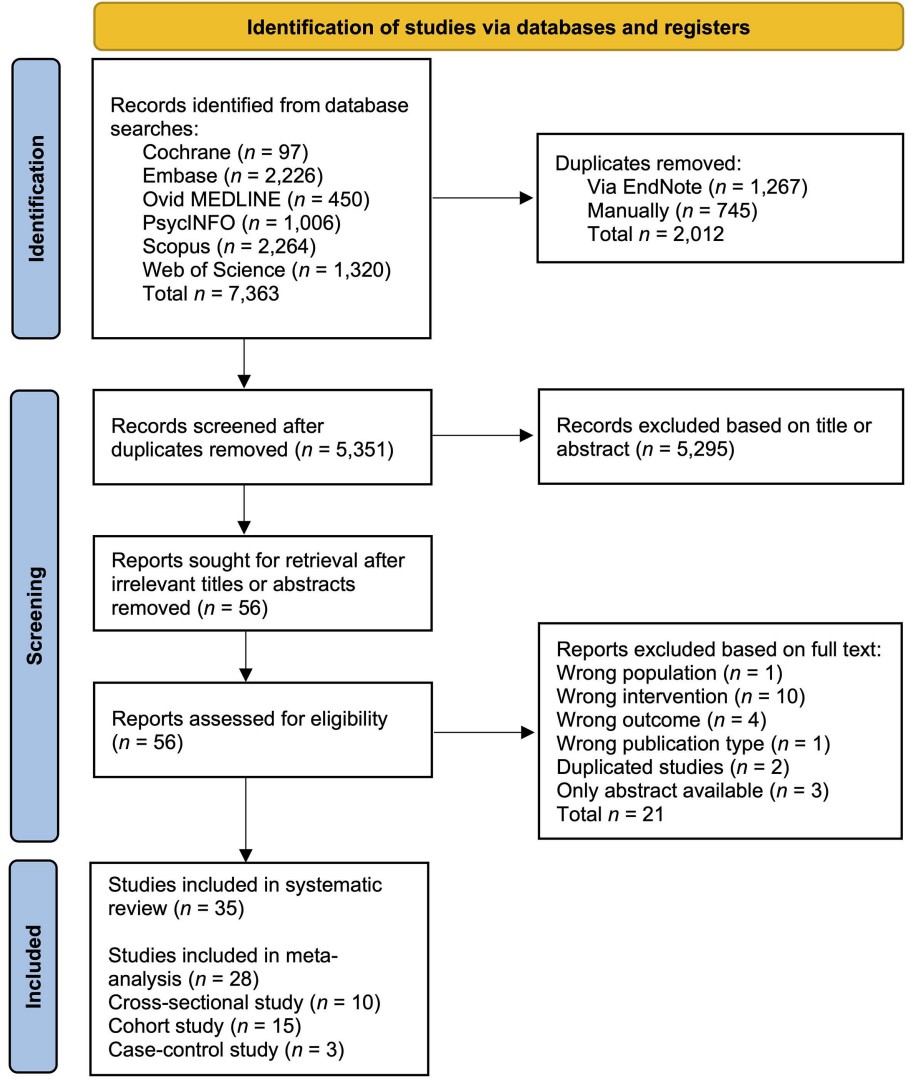

**Fig 1. Prisma flow diagram of study selection.**

dose–response meta-analysis (S2 Table). Interestingly, an average of 6 hours per day of TV watching was identified as the threshold for a statistically significant decrease in cognitive score (beta coefficient = −0.09 [95% CI: −0.17, −0.003]; seven studies; Fig 3A), and there was with a high degree of heterogeneity ($I^2$ = 71.80%, p = 0.002; S5 Table). All sensitivity analyses were consistent with the main findings. However, restricting the analysis to studies with a low-to-moderate risk of bias (five studies) or only to cohort studies (four studies) did not drastically change the degree of statistical heterogeneity (S6 Table).

Furthermore, our conventional meta-analysis revealed that increased TV-watching time was associated with a slight but significant decrease in cognitive score: the pooled mean difference was −0.02 (95% CI: −0.03, −0.003; six studies; Fig 3B), although the analyses showed a significant degree of heterogeneity ($I^2$ = 66.45%, p = 0.01). Notably, most sensitivity analyses yielded similar results, except when analyzing only cohort studies or only studies with a low-to-moderate risk of bias, where the association became null (Table 3). The scatter plot showing the relationship between TV-watching time and cognitive score is shown in S6 Fig.

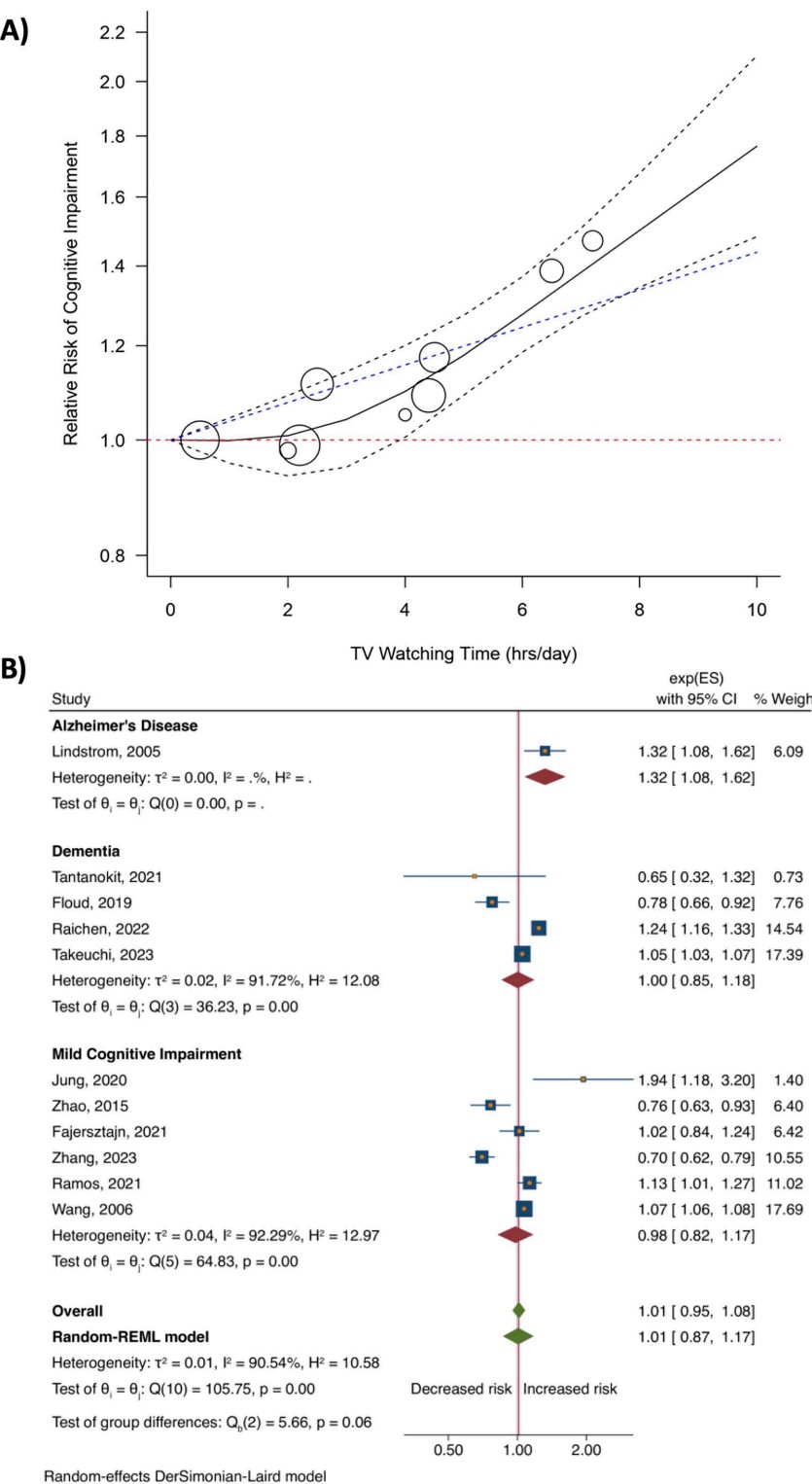

**Fig 2. The relationship between longer TV watching time and risk of cognitive impairment. A)** Dose-response meta-analysis of TV watching time (hours per day) and the risk of cognitive impairment based on 4 studies. **B)** Meta-analysis of a longer TV watching time, compared to a lower one, with the risk of cognitive impairment (11 studies). **Note:** The black dashed lines represent the 95% confidence interval, the blue dashed line represents the linear model, and the red dashed line represents the null value (RR = 1.00). The reference level is 0 hours per day.

**Table 2. Sensitivity and subgroup analysis of conventional meta-analysis of TV watching time and risk of cognitive impairment.**

| Analysis | Effect size (95% CI), p-value | I², p-value for heterogeneity |
|---|---|---|
| *Main analysis (n = 11)* | 1.01 (0.95, 1.08), 0.64 | 90.54%, < 0.001 |
| *Subgroup analysis* | | |
| By study design, | | |
| Cross-sectional (n = 2) | 1.16 (0.40, 3.37), 0.79 | 83.62%, 0.01 |
| Cohort (n = 6) | 1.00 (0.93, 1.07), 0.96 | 93.89%, < 0.001 |
| Case-control (n = 3) | 1.05 (0.79, 1.38), 0.75 | 87.89%, < 0.001 |
| By outcome | | |
| Alzheimer's disease (n = 1) | **1.32 (1.08, 1.62), 0.007** | NA |
| Dementia (n = 4) | 1.00 (0.85, 1.18), 0.97 | 91.72%, < 0.001 |
| MCI (n = 6) | 0.98 (0.82, 1.17), 0.82 | 92.29%, < 0.001 |
| By reported effect size | | |
| Hazard ratio (n = 4) | **1.07 (1.02, 1.13), 0.01** | 91.46%, < 0.001 |
| Odds ratio (n = 7) | 1.00 (0.79, 1.26), 0.99 | 89.58%, < 0.001 |
| By risk of bias | | |
| Low risk of bias (n = 5) | 0.96 (0.77, 1.21), 0.76 | 92.47%, < 0.001 |
| Moderate risk of bias (n = 6) | 1.03 (0.91, 1.18), 0.63 | 90.40%, < 0.001 |
| *Sensitivity analysis* | | |
| Random-REML model (n = 11) | 1.01 (0.87, 1.17), 0.91 | 98.93%, < 0.001 |
| Only adjusted effect size (n = 10) | 1.02 (0.96, 1.08), 0.57 | 91.34%, < 0.001 |
| Largest variance (n = 11) | 1.00 (0.94, 1.07), 0.91 | 90.89%, < 0.001 |
| Shorter follow-up (n = 11) | 1.01 (0.95, 1.08), 0.74 | 91.32%, < 0.001 |

## Publication bias and leave-one-out analyses

Visual inspection of the contour-enhanced and conventional funnel plots for cognitive-impairment risk (S7A & B Fig) suggested some asymmetry; however, Egger's test indicated no statistically significant publication bias (p-value = 0.43). Likewise, both the contour-enhanced funnel plot (S8A Fig) and the classical funnel plot (S8B Fig) of the cognitive score outcome also showed no apparent evidence of publication bias, with a p-value of 0.56 derived from Egger's test. Additionally, imputed results from the trim-and-fill analysis of the cognitive scores did not change our conclusions. In the leave-one-out analysis (S9 Fig), although most studies did not influence the findings, three studies might have dominated the main results. Omitting one study (Zhang et al., 2023 [66]) from the meta-analysis of cognitive impairment risk (S9A Fig) changed the results from null to significant (1.07 [95% CI: 1.01, 1.12]). In contrast, leaving out either of two studies (Shin et al., 2021 [63] and Maasakkers et al., 2021 [58]) in the meta-analysis of cognitive score reverted the results to null (S9B Fig).

Taken together, according to GRADE, our dose–response meta-analysis of cognitive impairment risk was rated as having a moderate level of certainty, whereas the dose–response meta-analysis of cognitive score had a low level of certainty, and the conventional meta-analyses of both cognitive impairment and cognitive scores had a very low level of certainty (S7 Table).

## Discussion

Our study is the first meta-analysis to explore the association between TV-watching time and cognitive outcomes in adults and older adults; it included 35 studies with a total of 1,292,052 participants. In the dose–response meta-analyses, we observed a nonlinear association between TV-watching time and unfavorable cognitive outcomes. Specifically, watching

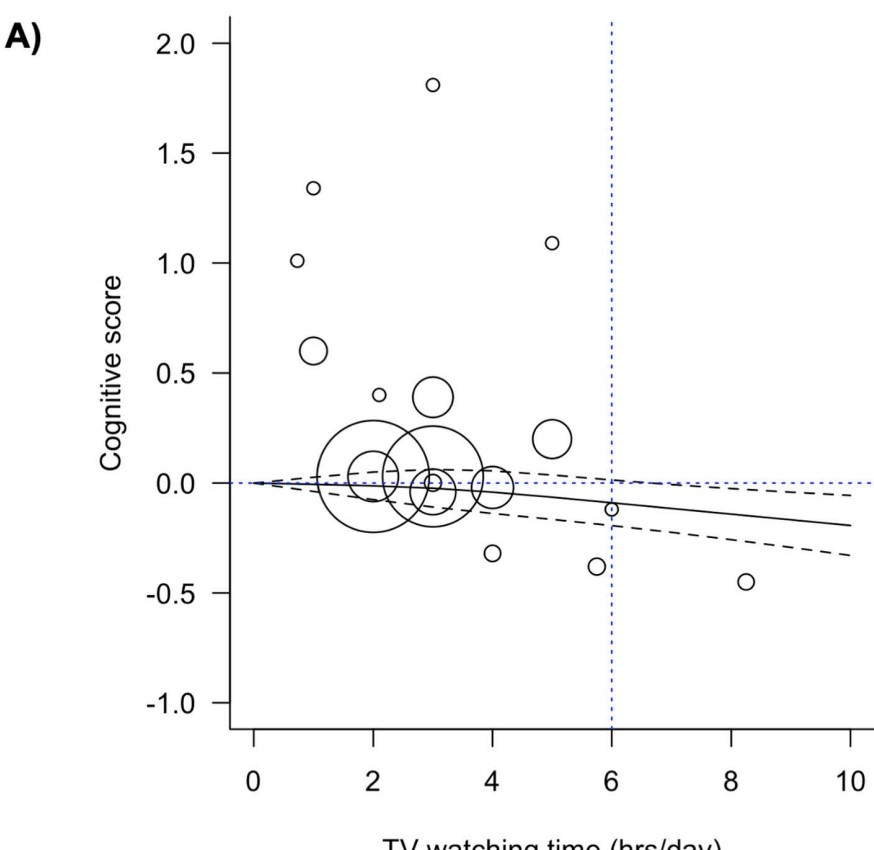

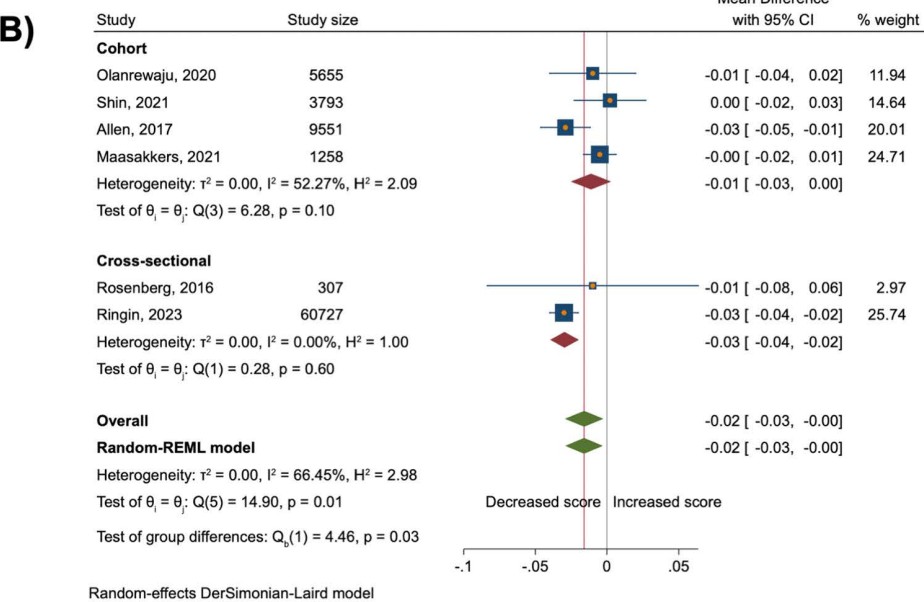

**Fig 3. The relationship between longer TV watching time and cognitive scores. A)** Dose-response meta-analysis of TV watching time (hours per day) and cognitive score fitted with restricted cubic spline with 3 knots (7 studies). **B)** Meta-analysis of a longer TV watching time, compared to a shorter one, with a cognitive score (6 studies).

**Table 3. Sensitivity and subgroup analysis between higher TV watching time and cognitive scores.**

| Analysis | Mean difference (95% CI), p-value | I², p-value for heterogeneity |
|---|---|---|
| *Main analysis (n = 6)* | **−0.02 (−0.03, −0.003), 0.02** | 66.45%, 0.01 |
| *Subgroup analysis* | | |
| By study design | | |
| Cross-sectional (n = 2) | **−0.03 (−0.04, −0.02), < 0.001** | 0.0%, 0.60 |
| Cohort (n = 4) | −0.01 (−0.03, 0.003), 0.12 | 52.27%, 0.10 |
| *Sensitivity analysis* | | |
| Included unweighted study (n = 7)* | **−0.02 (−0.03, −0.003), 0.02** | 60.12%, 0.02 |
| Low-to-moderate RoB (n = 5) | −0.01 (−0.03, 0.004), 0.13 | 69.86%, 0.01 |
| Largest SD (n = 6) | **−0.02 (−0.04, −0.001), 0.04** | 73.98%, 0.002 |
| Random-REML model (n = 6) | **−0.02 (−0.03, −0.003), 0.02** | 64.67%, 0.02 |
| Trim-and-fill analysis (n = 7)* | **−0.02 (−0.03, −0.01)** | NA |

**Note:** The main analysis was conducted using a DerSimonian-Laird random-effects model, and bold figures represent a statistically significant value (p-value < 0.05). *In the sensitivity analysis, which included an unweighted study, one extra study (Fajersztajn et al., 2021 [39]) was included in the analysis. In the trim-and-fill analysis, one ideal study was added to make the funnel plot more symmetrical. **Abbreviations:** CI, confidence interval; REML, restricted maximum likelihood; RoB, risk of bias; SD, standard deviation.

TV for ≥4 hours per day was associated with a significantly higher risk of cognitive impairment, while watching ≥6 hours per day was linked to lower cognitive scores. Although the conventional meta-analyses did not show an association between TV-watching time and the risk of cognitive impairment (except for an increased risk of AD), it has been shown that a longer TV-watching time is linked to a significantly lower cognitive score. These results support an association between TV watching and negative cognitive outcomes in adults and older adults.

## Association between TV watching and cognition

Through a dose–response meta-analysis, we identified a nonlinear association between TV-watching time and an increased risk of cognitive impairment, with a threshold of 4 hours per day. However, the results of our conventional meta-analysis did not demonstrate this association, which supports the nonlinear relationship, as this pattern cannot be captured by a conventional meta-analysis. Nevertheless, in subgroup analysis, the conventional meta-analysis provided important insights: a longer TV-watching time was associated with a significantly higher risk of AD, and when only hazard ratios were pooled. This points to the need for further studies for clarification because only one study was included in the analysis. Moreover, watching TV for ≥6 hours per day was associated with significantly lower cognitive scores. This was also supported by the results of our conventional meta-analysis. However, it must be noted that there was significant heterogeneity between studies regarding this association because of the different study designs.

In summary, we observed a significant nonlinear increase in the risk of cognitive impairment with TV-watching time and a significant decrease in cognitive scores after 6 hours. This difference might result from the difference in the nature of the two outcomes. The cognitive impairment data were binarized, whereas the cognitive scores were continuous. Moreover, there were differences in follow-up times and cognitive tests among studies.

## Implications of the association between TV watching and worsening cognition

Our findings are alarming because it has been demonstrated that adults may watch up to 7 hours of TV per day on average [69,70]. If the risk of cognitive impairment significantly increases at 4 hours of TV watching or more, an average adult watching 7 hours of TV per day would have a notably higher risk of cognitive impairment. This becomes even more critical

in the context of an aging population, because people tend to watch TV for longer as they get older [71,72]. The combination of increased TV-watching time with age and TV watching's association with a higher risk of cognitive impairment risk will undoubtedly increase the burden on the economy and public health system. Thus, it is imperative that alternative leisure activities are recommended for adults and older adults.

## Mechanisms linking TV watching to cognition

Evidence from the literature suggests that there is a direct association between TV watching and decreased brain volume in several parts of the brain, including parts related to language, memory, and communication that are usually affected by dementia [64,73]. This association persists even after adjusting for possible confounders such as physical activity [74], suggesting that there is a direct mechanism linking TV watching and cognitive impairment. Indirect effects of TV watching may also contribute to the association observed in this study. TV-watching time is often used as an indicator of how long a person is engaged in sedentary activity per day, and it has an inverse relationship with physical activity time [20,75]. Both TV watching and sedentary activity are associated with worse cognitive performance and cognitive impairment [76]. Furthermore, TV viewing is linked to diseases such as obesity and diabetes [20], as well as negative psychosocial outcomes such as loneliness, depression, and low life satisfaction, which could also increase dementia risk [77–79].

## Clinical implications

Although further studies are required to confirm the association, our study is the first meta-analysis to show a negative association between watching TV and cognitive outcomes. On an individual level, patients could be advised regarding the potential cognitive benefits of decreasing the time spent viewing TV because each hour of TV watching increases the risk of cognitive impairment. Additionally, regardless of whether TV watching is a causal factor in cognitive impairment, people (especially older adults) who spend most of their time watching TV may still benefit from monitoring and evaluation of cognitive impairment. Our findings could also inform public health strategies to dissuade adults and older adults from watching TV to improve their cognitive health, and they emphasize the need for adults to engage in other cognitive and daily leisure activities; public health providers could use this information to devise policies aimed at improving cognitive health. Since this study focuses on the relationship between TV watching (a commonly used marker for sedentary behavior) and cognition, it may be possible to apply the results from this study to similar sedentary activities, such as watching internet videos and using streaming platforms. However, it is important to note the different variables associated with each type of sedentary activity, for example, some activities may be more appealing to certain age and gender groups than others. In addition, some activities, such as watching internet videos, might be associated with a higher degree of interaction with the user than watching television. Factors such as these may alter the relationship between different sedentary behavior and cognition [40]. Thus, the results from this study must be extrapolated with caution.

## Strengths and limitations

To our best knowledge, this is the first study to comprehensively review and meta-analyze the associations of TV-watching time with cognitive scores and cognitive impairment risk. Furthermore, we performed a dose–response meta-analysis to potentially identify a nonlinear trend that may not be captured by the conventional analytic approach. However, there are some noteworthy limitations to this study. First, all the included studies were observational in design, meaning that several alternative explanations—particularly the influence of residual confounders and reverse causality (e.g., individuals in the subclinical stage of cognitive impairment may spend most of their time watching TV due to physical limitations)—cannot be ruled out. Therefore, causality cannot be inferred from our findings. Nonetheless, reverse causality is not a major concern in our study. This is because the dose–response meta-analysis findings on time spent watching TV and the risk of cognitive impairment are based solely on cohort studies, which are less prone to reverse causality compared to cross-sectional studies (S3 Table). Additionally, most of the included cohort studies (66.7%) were assessed as having a low risk of reverse causality in one

domain of the NOS. Furthermore, the subgroup analysis of the dose-response meta-analysis for cognitive scores based on cohort studies yielded results consistent with the main analysis (S5 Table). Second, some results showed a significant degree of heterogeneity. Consequently, for some findings, the certainty of the evidence was rated as low to very low according to GRADE, such that readers should exercise caution when interpreting the findings. Our subgroup and sensitivity analyses (Table 2) suggest that study design, outcome type, reported effect sizes, and risk of bias were not the primary contributors to the heterogeneity. This persistent heterogeneity across subgroups suggests that other factors, such as unmeasured confounders, may contribute. These may include differences in TV assessment methods (e.g., self-reported hours vs. categorical measures), regional variations in viewing habits, and the use of different cognitive measures and scales (e.g., Mini Mental State Exam (MMSE) vs. Montreal Cognitive Assessment (MoCA) vs self-report). These factors likely contributed to variability beyond study design or risk of bias; thus, the pooled estimates should be interpreted with caution. Interestingly, in the leave-one-out analysis, one study by Zhang et al. [66] appears to be a potential influential source. This is because the exclusion of the study shifts the pooled estimate, suggesting it contributes to observed heterogeneity. The apparent funnel-plot asymmetry is more plausibly driven by between-study heterogeneity and variation in study precision than by small-study publication bias. In addition, our comprehensive search strategy—covering six major databases (Cochrane, Ovid MEDLINE, Ovid Embase, PsycINFO, Scopus, and Web of Science)—reduces the likelihood that relevant studies were missed, further minimizing the chance that publication bias explains the pattern observed. Third, this study focused on adults and older adults, so its findings may not be applicable to younger populations (< 18 years old). Fourth, it should be noted that forest plots with a limited number of studies (e.g., subgroup analyses with ≤2 studies) as shown in Fig 3B should be interpreted with caution. P-values may not reliably indicate true between-group differences under these conditions. Lastly, although we used fully adjusted estimates where available, residual confounding remains a key limitation due to variability in covariates across studies and potential unmeasured factors such as social engagement, depression, or baseline cognitive status.

## Conclusion and future directions

Determining the relationship between the most popular leisure activity among adults and older adults, TV watching, and cognitive outcomes has never been more important because the world is heading towards an aging population crisis. The current evidence supports an association between longer TV-watching time and negative cognitive outcomes in adults and older adults; however, causality in the relationship remains to be fully elucidated. Additionally, future studies should consider the relationship between different types of TV programming on cognitive decline in adults as there is currently a lack of evidence on this specific topic. Nevertheless, this study has established that the answer to the question of how long one can spend watching TV per day without hindering cognitive performance is less than 4–6 hours in adults and older adults. The findings of this study could be used as a basis for public advice pertaining to healthier aging.

## Supporting information

**S1 Fig. ROBVIS: Risk-of-bias VISualization for cross-sectional studies.** (A) Traffic Light Plot for risk of bias domains. (B) Weighted bar plots of the distribution of risk-of-bias for each domain.
(DOCX)

**S2 Fig. Risk-of-bias for cohort studies.** (A) Traffic Light Plot for risk of bias domains. (B) Weighted bar plots of the distribution of risk-of-bias for each domain.
(DOCX)

**S3 Fig. Risk-of-bias for case-control studies.** (A) Traffic Light Plot for risk of bias domains. (B) Weighted bar plots of the distribution of risk-of-bias for each domain.
(DOCX)

**S4 Fig. Scatter plot of TV watching time (dose; x) and cognitive impairment risk (logrr; y) (4 studies).** Each circle depicts the logrr and inver_se of cognitive impairment risk at each dose of TV watching time reported in each study.
(DOCX)

**S5 Fig. Subgroup Meta-Analysis of TV Watching Time and Cognitive Impairment Risk by Outcome and Study Design.** Conventional meta-analysis of higher versus lower TV watching time and the associated risk of cognitive impairment (11 studies), with subgroup analyses by outcome (upper panel) and study design (lower panel).
(DOCX)

**S6 Fig. Scatter Plot of TV Watching Time (dose; x) and the Change in Cognitive Score (beta coefficient; y).** Each circle depicts the beta coefficient and inver_se of the change in cognitive score at each dose of TV watching time reported in each study.
(DOCX)

**S7 Fig. Funnel Plot Analyses for Publication Bias in the Association Between TV Watching Time and Cognitive Impairment Risk.** (A) Contour-enhanced funnel plot and (B) conventional funnel plot assessing publication bias in the association between TV watching time and risk of cognitive impairment (11 studies).
(DOCX)

**S8 Fig. Funnel Plot Analyses for Publication Bias in the Association Between TV Watching Time and Cognitive Performance Score.** (A) Contour-enhanced funnel plot and (B) conventional funnel plot assessing publication bias in the association between TV watching time and cognitive score (6 studies).
(PDF)

**S9 Fig. Leave-One-Out Sensitivity Analyses for the Association Between TV Watching Time and Cognitive Outcomes.** Leave-one-out analysis evaluating the influence of each individual study on the pooled estimate of the association between TV watching time and (A) risk of cognitive impairment (11 studies) and (B) cognitive score (6 studies).
(DOCX)

**S1 Table. PRISMA checklist.**
(DOCX)

**S2 Table. Information criteria of each dose-response meta-analysis model.**
(DOCX)

**S3 Table. Predicted relative risk of cognitive impairment based on dose-response meta-analysis model (n = 4).**
(DOCX)

**S4 Table. Subgroup analysis of TV watching time and risk of cognitive impairment according to reported effect sizes.**
(DOCX)

**S5 Table. Predicted cognitive score based on dose-response meta-analysis model (n = 7).**
(DOCX)

**S6 Table. Sensitivity analysis of average TV watching time and predicted cognitive score.**
(DOCX)

**S7 Table. Certainty of findings according to GRADE.**
(DOCX)

**S1 File. Supplementary methods.**
(DOCX)

## Acknowledgments

All tools and facilities were supported by Chulabhorn Royal Academy and University of Phayao. We also thank Michael Irvine, PhD, from Edanz (www.edanz.com/ac) for editing a draft of this manuscript.

## Author contributions

**Conceptualization:** Hattapark Dejakaisaya, Chanawee Hirunpattarasilp.

**Data curation:** Hattapark Dejakaisaya, Wiriya Mahikul, Nat Na-Ek, Chanawee Hirunpattarasilp.

**Formal analysis:** Hattapark Dejakaisaya, Wiriya Mahikul, Nat Na-Ek, Chanawee Hirunpattarasilp.

**Funding acquisition:** Hattapark Dejakaisaya, Chanawee Hirunpattarasilp.

**Investigation:** Hattapark Dejakaisaya, Wiriya Mahikul, Nat Na-Ek, Chanawee Hirunpattarasilp.

**Methodology:** Hattapark Dejakaisaya, Wiriya Mahikul, Nat Na-Ek, Chanawee Hirunpattarasilp.

**Project administration:** Hattapark Dejakaisaya, Chanawee Hirunpattarasilp.

**Resources:** Hattapark Dejakaisaya, Chanawee Hirunpattarasilp.

**Software:** Hattapark Dejakaisaya, Wiriya Mahikul, Chanawee Hirunpattarasilp.

**Supervision:** Hattapark Dejakaisaya, Chanawee Hirunpattarasilp.

**Validation:** Hattapark Dejakaisaya, Wiriya Mahikul, Nat Na-Ek, Chanawee Hirunpattarasilp.

**Visualization:** Hattapark Dejakaisaya, Wiriya Mahikul, Nat Na-Ek, Chanawee Hirunpattarasilp.

**Writing – original draft:** Hattapark Dejakaisaya, Wiriya Mahikul, Nat Na-Ek, Chanawee Hirunpattarasilp.

**Writing – review & editing:** Hattapark Dejakaisaya, Wiriya Mahikul, Nat Na-Ek, Chanawee Hirunpattarasilp.

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
