## [Decision Letter · Decision Letter 0]

5 Jun 2025

PONE-D-25-04250Television watching and cognitive outcomes in adults and older adults: A systematic review and dose–response meta-analysis of observational studiesPLOS ONE

Dear Dr. Chanawee Hirunpattarasilp,

Thank you for submitting your manuscript to PLOS ONE. We would like to apologize for the delay in response. After careful consideration, we feel that it has merit but does not fully meet PLOS ONE’s publication criteria as it currently stands. Therefore, we invite you to submit a revised version of the manuscript that addresses the points raised during the review process.

Please submit your revised manuscript by Jul 20 2025 11:59PM**.** If you will need more time than this to complete your revisions, please reply to this message or contact the journal office at plosone@plos.org . Please include the following items when submitting your revised manuscript:

We look forward to receiving your revised manuscript.

Kind regards,

Anat Rotstein, PhD

Academic Editor

PLOS ONE

“This research was supported by Chulabhorn Royal Academy and University of Phayao and Thailand Science Research and Innovation Fund (Fundamental Fund 2024).”

“This research was supported by Chulabhorn Royal Academy and University of Phayao and Thailand Science Research and Innovation Fund (Fundamental Fund 2024). We thank Michael Irvine, PhD, from Edanz (www.edanz.com/ac) for editing a draft of this manuscript.”

“This research was supported by Chulabhorn Royal Academy and University of Phayao and Thailand Science Research and Innovation Fund (Fundamental Fund 2024).

Reviewers' comments:

Reviewer's Responses to Questions

**Comments to the Author**

1. Is the manuscript technically sound, and do the data support the conclusions?

Reviewer #1: No

Reviewer #2: Yes

2. Has the statistical analysis been performed appropriately and rigorously? 

Reviewer #1: No

Reviewer #2: Yes

3. Have the authors made all data underlying the findings in their manuscript fully available?

Reviewer #1: Yes

Reviewer #2: Yes

4. Is the manuscript presented in an intelligible fashion and written in standard English?

Reviewer #1: Yes

Reviewer #2: Yes

5. Review Comments to the Author

Reviewer #1: The paper is interesting. However, there are some concerns from a statistical perspective.

Certainty of findings grade in supplement were all very low to moderate concerning evidence of certainty which may indicate that the association sought by the investigators is not very strong.

All the included studies were observational in design, meaning that the authors had doubt of the associations examined as they note, “particularly the influence of residual confounders and reverse causality (e.g., individuals in the subclinical stage of cognitive impairment may spend most of their time watching TV due to physical limitations)—cannot be ruled out. Therefore, causality cannot be inferred from our findings.” This use of the term , causality, by the authors is inappropriate as the effect size measures in this study and most statistical studies involves evidence of association and not causality. Thus causality is not in play, especially in a meta-analysis, which is primarily exploratory or suggestive. The authors note correctly themselves that regarding the association between TV-watching time and cognitive score, they found a nonlinear relationship via a three-knot restricted cubic spline model, which was the best-fitting model in the dose–response meta-analysis according to the AIC and BIC in Table 3 of the supplement. Likewise, they give the same quantitative interpretation correctly for the Analysis of the risk of cognitive impairment via the term association. That is as far as they can go in the interpretation. They cannot assess causality with their approach. They should rethink this interpretation and stay with association, strong or weak.

In general the synthesis of the findings and its organization was done and interpreted correctly including the risk of bias assessment.

The analysis requires much clarification. The authors note in the analysis portion that, where possible, fully adjusted effect sizes were used. This is not clear . What adjustment? There is reference to confounders and causes for a weak or strong association in some cases. However, there is no real detailed evidence of relevant confounder discussion (clinical or demographic in addition to the self report limitations noted by the authors) in the text or supplement. Also, the I squares are given with p-values in the Figures 2 and 3 or in the supplement with no explanation of the causes of significance , if any. The funnel plots are well done and presented in the supplement. However, the asymmetry (Figure 7A) should be explained. Also, the authors should explain in more detail the difference in the graphical presentations of Supplementary Figures 6 and 7 vs. Supplementary Figures 8 and 9.

Also, there should be a caution about few cross sectional studies on the Forest plots (Figure 3B) and care in interpreting the p-values when there are so few studies on a Forest plot. The entire document should be edited to be sure relevant detail is provided to the reader.

Reviewer #2: The manuscript presents a systematic review and meta-analysis to investigate the association between television watching and cognitive outcomes in adults and older adults. The topic is of significant relevance, the PRISMA checklist is followed, and the supplement is comprehensive. The inclusion of dose–response analyses and a large data set are major strengths. To improve transparency, reproducibility, and reader accessibility, I have several suggestions as detailed below. Overall, this is a valuable and important manuscript that will benefit from greater clarity and integration of its supplementary material.

Methods

- Data Extraction: Please summarize the outcome selection hierarchy and its rationale in the main Methods for clarity. Consider moving a summary of the main variables extracted (as per the supplement) to the main text.

- Assessment of Bias: Add a brief explanation of the NOS domains so that readers unfamiliar with the tool can understand your assessment. For transparency, summarize the distribution of risk categories (low, moderate, high risk of bias) in the main text.

- Statistical Analysis: Clearly summarize the hierarchy and decision process for handling multiple outcomes in the main Methods.

Results

- Please move a summary table of study characteristics into the main Results for quick reader reference.

- Given the high heterogeneity in many analyses, please provide a more granular exploration or discussion of possible sources (e.g., differences in TV assessment methods, region, cognitive test used).

- Review, update, and ensure consistency for all figure/table numbers and legends so that all are referenced appropriately in the text.

- Clarify and standardize decimal place reporting.

- Use color schemes/line types in figures that are clear to all readers.

- The main text cites references in mixed formats; please standardize to one citation style throughout.

- Key dose–response, subgroup, and sensitivity findings should be highlighted within the main Results section, not just in the supplement.

Discussion

- The discussion focuses solely on "television" as a sedentary behavior. Please also discuss whether findings may generalize to other screen-based activities (e.g., internet videos, streaming platforms) and address the limitations of this extrapolation.

- Briefly consider whether the content of TV programming might matter (e.g., educational vs. non-educational, passive vs. active).

---

## [Author Response · Author response to Decision Letter 1]

27 Jun 2025

Response to Editor

PONE-D-25-04250

Television watching and cognitive outcomes in adults and older adults: A systematic review and dose–response meta-analysis of observational studies

PLOS ONE

We have ensured that the manuscript meets PLOS ONE’s style requirements.

“This research was supported by Chulabhorn Royal Academy and University of Phayao and Thailand Science Research and Innovation Fund (Fundamental Fund 2024).”

We have included the amended Funding Statement within the cover letter as below:

“This research was supported by Chulabhorn Royal Academy and University of Phayao and Thailand Science Research and Innovation Fund (Fundamental Fund 2024). There was no additional external funding received for this study.”

ORCID iD has been updated.

“This research was supported by Chulabhorn Royal Academy and University of Phayao and Thailand Science Research and Innovation Fund (Fundamental Fund 2024). We thank Michael Irvine, PhD, from Edanz (www.edanz.com/ac) for editing a draft of this manuscript.”

“This research was supported by Chulabhorn Royal Academy and University of Phayao and Thailand Science Research and Innovation Fund (Fundamental Fund 2024).

Please see comment number 2 for the updated Funding Statement. The Acknowledgement now reads as:

“All tools and facilities were supported by Chulabhorn Royal Academy and University of Phayao. We also thank Michael Irvine, PhD, from Edanz (www.edanz.com/ac) for editing a draft of this manuscript.”

The ethics statement has been stated in the Materials and methods on page 4, para 3 as follows:

“Further, this project received an ethics exemption from Chulabhorn Royal Academy’s ethics committee (project number EC 052/2566).” 

We have now included the captions for Supporting Information files at the end of the manuscript.

We have uploaded the figure files to the PACE and ensured that the figures meet PLOS requirements.

Response to Reviewers

PONE-D-25-04250

Television watching and cognitive outcomes in adults and older adults: A systematic review and dose–response meta-analysis of observational studies

PLOS ONE

1. Reviewer #1: The paper is interesting. However, there are some concerns from a statistical perspective. Certainty of findings grade in supplement were all very low to moderate concerning evidence of certainty which may indicate that the association sought by the investigators is not very strong.

1.1 All the included studies were observational in design, meaning that the authors had doubt of the associations examined as they note, “particularly the influence of residual confounders and reverse causality (e.g., individuals in the subclinical stage of cognitive impairment may spend most of their time watching TV due to physical limitations)—cannot be ruled out. Therefore, causality cannot be inferred from our findings.” This use of the term , causality, by the authors is inappropriate as the effect size measures in this study and most statistical studies involves evidence of association and not causality. Thus causality is not in play, especially in a meta-analysis, which is primarily exploratory or suggestive. The authors note correctly themselves that regarding the association between TV-watching time and cognitive score, they found a nonlinear relationship via a three-knot restricted cubic spline model, which was the best-fitting model in the dose–response meta-analysis according to the AIC and BIC in Table 3 of the supplement. Likewise, they give the same quantitative interpretation correctly for the Analysis of the risk of cognitive impairment via the term association. That is as far as they can go in the interpretation. They cannot assess causality with their approach. They should rethink this interpretation and stay with association, strong or weak.

We thank the reviewer for the comment. We acknowledge the limitations of observational studies and agree that our findings can only indicate associations, not causality. In response to the reviewer’s concern, we have revised the manuscript to remove or rephrase any mention of "causality" to ensure that the interpretation of our results remains within the appropriate scope of observational research. Specifically, we now consistently use the term “association” throughout the manuscript to reflect the nature of the evidence.

1.2 In general the synthesis of the findings and its organization was done and interpreted correctly including the risk of bias assessment.

We thank the reviewer for the positive comment.

1.3. The analysis requires much clarification. The authors note in the analysis portion that, where possible, fully adjusted effect sizes were used. This is not clear . What adjustment? There is reference to confounders and causes for a weak or strong association in some cases. However, there is no real detailed evidence of relevant confounder discussion (clinical or demographic in addition to the self report limitations noted by the authors) in the text or supplement.

To improve clarity, we have now explained the logic behind using the fully adjusted effect sizes (page 8, para 2) and listed the covariates adjusted for in each included study in Table 1:

“Where possible, we used the effect sizes from models that included the most comprehensive set of covariates reported in each primary study to account for potential cofounders such as age, sex, education, socioeconomic status, lifestyle factors (e.g. physical activity, smoking), and other comorbidities. These factors, such as older age, female sex and lower education attainment, can negatively affect the cognitive outcomes [40]. Details of covariate adjusted for in each study are provided in Table 1.”

1.4. Also, the I squares are given with p-values in the Figures 2 and 3 or in the supplement with no explanation of the causes of significance , if any. The funnel plots are well done and presented in the supplement. However, the asymmetry (Figure 7A) should be explained. Also, the authors should explain in more detail the difference in the graphical presentations of Supplementary Figures 6 and 7 vs. Supplementary Figures 8 and 9.

We have now added the following text to discuss the significant heterogeneity, suggested by I² and its p-values, on page 27, para 1:

“Second, some results showed a significant degree of heterogeneity. Consequently, for some findings, the certainty of the evidence was rated as low to very low according to GRADE, such that readers should exercise caution when interpreting the findings. Our subgroup and sensitivity analyses (Table 2) suggest that study design, outcome type, reported effect sizes, and risk of bias were not the primary contributors to the heterogeneity. This persistent heterogeneity across subgroups suggests that other factors, such as unmeasured confounders, may contribute. These may include differences in TV assessment methods (e.g., self-reported hours vs. categorical measures), regional variations in viewing habits, and the use of different cognitive measures and scales (e.g., MMSE vs. MoCA vs self-report). These factors likely contributed to variability beyond study design or risk of bias; thus, the pooled estimates should be interpreted with caution. Interestingly, in the leave-one-out analysis, (Zhang et al., 2023 [66]) appears to be a potential influential source, as its exclusion shifts the pooled estimate, suggesting it contributes to observed heterogeneity.”

Regarding the asymmetry of the funnel plot (Supplementary Figure 7), Egger’s test indicated no statistically significant publication bias (p-value = 0.43). The asymmetry is more likely to be driven by between-study heterogeneity and variation in study precision.

The Egger’s test result is stated on page 23, para 1 as follows:

“Visual inspection of the contour-enhanced and conventional funnel plots for cognitive-impairment risk (S7A & B Fig) suggested some asymmetry; however, Egger’s test indicated no statistically significant publication bias (p-value = 0.43).”

The cause of asymmetry is discussed on page 28, para 1:

“The apparent funnel-plot asymmetry is more plausibly driven by between-study heterogeneity and variation in study precision than by small-study publication bias. In addition, our comprehensive search strategy—covering six major databases (Cochrane, Ovid MEDLINE, Ovid Embase, PsycINFO, Scopus, and Web of Science)—reduces the likelihood that relevant studies were missed, further minimizing the chance that publication bias explains the pattern observed.”

The graphical differences between Supplementary Figures 6-9 reflect the different outcomes assessed. We have now changed the figure legend for Supplementary Figure 6 to further explain the graph as follows:

“S6 Fig. Scatter Plot of TV Watching Time (dose; x) and the Change in Cognitive Score (beta coefficient; y). Each circle depicts the beta coefficient and inver_se of the change in cognitive score at each dose of TV watching time reported in each study.”

Supplementary Figures 7 and 8 showed funnel plots with and without contour enhancements to evaluate potential publication bias for cognitive impairment risk and cognitive scores, respectively. The difference is stated on page 23, para 1 as follows:

“Visual inspection of the contour-enhanced and conventional funnel plots for cognitive-impairment risk (S7A & B Fig) suggested some asymmetry; however, Egger’s test indicated no statistically significant publication bias (p-value = 0.43). Likewise, both the contour-enhanced funnel plot (S8A Fig) and the classical funnel plot (S8B Fig) of the cognitive score outcome also showed no apparent evidence of publication bias, with a p-value of 0.56 derived from Egger’s test.”

Supplementary Figure 9 shows leave-one-out sensitivity analyses for both outcomes, illustrating the influence of individual studies on the overall pooled estimates. The leave-one-out analysis is explained on page 9, para 2:

“Additionally, the influence of each study was examined by performing a leave-one-out analysis.”

And the result of the Supplementary Figure 9 has been discussed on page 23, para 1:

“In the leave-one-out analysis (S9 Fig), although most studies did not influence the findings, three studies might have dominated the main results. Omitting one study (Zhang et al., 2023 [66]) from the meta-analysis of cognitive impairment risk (S9A Fig) changed the results from null to significant (1.07 [95% CI: 1.01, 1.12]). In contrast, leaving out either of two studies (Shin et al., 2021 [63] and Maasakkers et al., 2021 [58]) in the meta-analysis of cognitive score reverted the results to null (S9B Fig).“

1.5. Also, there should be a caution about few cross sectional studies on the Forest plots (Figure 3B) and care in interpreting the p-values when there are so few studies on a Forest plot. The entire document should be edited to be sure relevant detail is provided to the reader.

We agree with the reviewer. We have added the following sentence to the limitation section in our discussion (page 28, para 1):

“Fourth, it should be noted that forest plots with a limited number of studies (e.g., subgroup analyses with ≤2 studies) as shown in Fig 3B should be interpreted with caution. P-values may not reliably indicate true between-group differences under these conditions.”

2. Reviewer #2: The manuscript presents a systematic review and meta-analysis to investigate the association between television watching and cognitive outcomes in adults and older adults. The topic is of significant relevance, the PRISMA checklist is followed, and the supplement is comprehensive. The inclusion of dose–response analyses and a large data set are major strengths. To improve transparency, reproducibility, and reader accessibility, I have several suggestions as detailed below. Overall, this is a valuable and important manuscript that will benefit from greater clarity and integration of its supplementary material.

2.1 Methods

2.1.1 Data Extraction: Please summarize the outcome selection hierarchy and its rationale in the main Methods for clarity. Consider moving a summary of the main variables extracted (as per the supplement) to the main text.

Thank you for your feedback. As per your request, we have moved the following summary of the main variables extracted from the supplementary file to the data extraction section in the main text (page 6, para 2):

“Studies with overlapping populations or from the same database were ranked based on a designed hierarchy and the studies with the highest hierarchical score were included. Briefly, studies were ranked based on 1) the most relevant outcomes (e.g. dementia, cognitive impairment, and cognitive score); 2) sample size (largest

---

## [Decision Letter · Decision Letter 1]

20 Aug 2025

PONE-D-25-04250R1Television watching and cognitive outcomes in adults and older adults: A systematic review and dose-response meta-analysis of observational studiesPLOS ONE

Dear Dr. Hirunpattarasilp,

Thank you for submitting your manuscript to PLOS ONE. After careful consideration, we feel that it has merit but does not fully meet PLOS ONE’s publication criteria as it currently stands. Therefore, we invite you to submit a revised version of the manuscript that addresses the points raised during the review process.

We look forward to receiving your revised manuscript.

Kind regards,

Anat Rotstein, PhD

Academic Editor

PLOS ONE

**Journal Requirements:**

Reviewers' comments:

Reviewer's Responses to Questions

**Comments to the Author**

1. If the authors have adequately addressed your comments raised in a previous round of review and you feel that this manuscript is now acceptable for publication, you may indicate that here to bypass the “Comments to the Author” section, enter your conflict of interest statement in the “Confidential to Editor” section, and submit your "Accept" recommendation.

Reviewer #1: All comments have been addressed

Reviewer #2: All comments have been addressed

2. Is the manuscript technically sound, and do the data support the conclusions?

Reviewer #1: (No Response)

Reviewer #2: Yes

3. Has the statistical analysis been performed appropriately and rigorously? 

Reviewer #1: (No Response)

Reviewer #2: Yes

4. Have the authors made all data underlying the findings in their manuscript fully available?

Reviewer #1: (No Response)

Reviewer #2: Yes

5. Is the manuscript presented in an intelligible fashion and written in standard English?

Reviewer #1: (No Response)

Reviewer #2: No

6. Review Comments to the Author

**Reviewer #1:**  Comments have been addressed. Please do not associate significance with a result unless it is a statistically significant result. For example "Additionally, watching ≥6 35 hours of television per day was associated with a significant decrease in cognitive score (standardized 36 beta coefficient = -0.09; 95% CI: -0.17, 0.003; I² = 71.8%; seven studies)." The CI here includes zero and the result is not significant. Please adjust accordingly and other edits as may be needed.

**Reviewer #2: ** Thank you for sharing this new version of the manuscript. There are still a few minor comments I would like to highlight:

- Review the numbering of figures. For example, there are two “Fig 1” referenced in the text (e.g., "Fig 1. Prisma flow diagram of study selection" and "Fig 1. The relationship between longer TV watching time and risk of cognitive impairment").

- Please review the text for consistency in the number of studies and participants included in the systematic review. The abstract and results report the same numbers, but the discussion gives different figures. Abstract/Results: “35 studies with 1,292,052 participants…” Discussion: “34 studies with a total of 1,246,876…”

- Please spell out abbreviations such as MMSE and MoCA at first mention in the text.

7. PLOS authors have the option to publish the peer review history of their article (what does this mean? ). If published, this will include your full peer review and any attached files.

**Do you want your identity to be public for this peer review?** For information about this choice, including consent withdrawal, please see our Privacy Policy .

Reviewer #1: No

Reviewer #2: No

---

## [Author Response · Author response to Decision Letter 2]

23 Aug 2025

Response to Reviewers

PONE-D-25-04250R1

Television watching and cognitive outcomes in adults and older adults: A systematic review and dose–response meta-analysis of observational studies

PLOS ONE

1. Reviewer #1: Comments have been addressed. Please do not associate significance with a result unless it is a statistically significant result. For example "Additionally, watching ≥6 35 hours of television per day was associated with a significant decrease in cognitive score (standardized 36 beta coefficient = -0.09; 95% CI: -0.17, 0.003; I² = 71.8%; seven studies)." The CI here includes zero and the result is not significant. Please adjust accordingly and other edits as may be needed.

Thank you for carefully reviewing our manuscript and for providing valuable comments to improve it. Upon further examination, we found that the confidence interval had been reported incorrectly: the minus sign was inadvertently omitted. It should read –0.17 to –0.003, rather than –0.17 to 0.003, as confirmed in the Results section (page 21, line 261) and Supplementary Table 5. We have also reviewed the entire manuscript to ensure the accuracy of all reported results.

2. Reviewer #2: Thank you for sharing this new version of the manuscript. There are still a few minor comments I would like to highlight:

2.1 Review the numbering of figures. For example, there are two “Fig 1” referenced in the text (e.g., "Fig 1. Prisma flow diagram of study selection" and "Fig 1. The relationship between longer TV watching time and risk of cognitive impairment").

Thank you for this input. We have reviewed the numbering of figures as requested and ensure that all the figures referenced in the text are accurate.

2.2 Please review the text for consistency in the number of studies and participants included in the systematic review. The abstract and results report the same numbers, but the discussion gives different figures. Abstract/Results: “35 studies with 1,292,052 participants…” Discussion: “34 studies with a total of 1,246,876…”

Thank you for identifying this discrepancy between the two sentences. We have made the necessary changes across the manuscript to ensure the consistency of number of studies included. Specifically, we changed the number in the discussion to match the number from abstract and results. The sentence now read “Our study is the first meta-analysis to explore the association between TV-watching time and cognitive outcomes in adults and older adults; it included 35 studies with a total of 1,292,052 participants.” (page 24, line 301).

2.3 Please spell out abbreviations such as MMSE and MoCA at first mention in the text.

Thank you for your valuable feedback. In addition to the list of abbreviations at the footnote of Table 1, we have now spelled out abbreviations such as MMSE and MoCA in the “Strengths and Limitations” section (page 27, line 393) of the “discussion”. The sentence now read “These may include differences in TV assessment methods (e.g., self-reported hours vs. categorical measures), regional variations in viewing habits, and the use of different cognitive measures and scales (e.g., Mini Mental State Exam (MMSE) vs. Montreal Cognitive Assessment (MoCA) vs self-report).”.

---

## [Editor Report · Decision Letter 2]

1 Sep 2025

Television watching and cognitive outcomes in adults and older adults: A systematic review and dose-response meta-analysis of observational studies

PONE-D-25-04250R2

Dear Dr. Chanawee Hirunpattarasilp,

We’re pleased to inform you that your manuscript has been judged scientifically suitable for publication and will be formally accepted for publication once it meets all outstanding technical requirements.

**Please make the following minor edits when proofing:**

1. Please delete the sentence beginning in line 103 ending in line 104 as it is repeated twice ("Two reviewers (HD, CH) extracted the relevant information independently, and disagreements

were resolved by discussion, or by a third reviewer (WM, NN) if necessary").

2. Line 375, please use analytic instead of analysis ("Furthermore, we performed a dose–response meta-analysis to potentially identify a nonlinear trend that may not be captured by the conventional ANALYTIC approach.")

3. Rephrase the sentence starting in mid-line 398 for clarity (Interestingly, in the leave-one-out analysis, (Zhang et al., 2023 [66]) appears to be a potential influential source, as its exclusion shifts the pooled estimate, suggesting it contributes to observed heterogeneity.)

Kind regards,

Anat Rotstein, PhD

Academic Editor

PLOS ONE

---

## [Editor Report · Acceptance letter]

PONE-D-25-04250R2

PLOS ONE

Dear Dr. Hirunpattarasilp,

I'm pleased to inform you that your manuscript has been deemed suitable for publication in PLOS ONE. Congratulations! Your manuscript is now being handed over to our production team.

Kind regards,

on behalf of

Dr. Anat Rotstein

Academic Editor

PLOS ONE